# Spatial Patterns of Production-Distribution-Consumption Cycle: The Specifics of Developing Russia

**Venera Timiryanova** [1,*], **Konstantin Grishin** [1] **and Dina Krasnoselskaya** [2]

1   Institute of Economics, Finance and Business, Bashkir State University, Zaki Validi St., 32, 450076 Ufa, Russia; grishin2472@yandex.ru
2   Institute of Economics and Service, Ufa State Petroleum Technological University, Cosmonauts st., 1, 450064 Ufa, Russia; dina-hamzina@mail.ru
*   Correspondence: veneratimiryanova@mail.ru; Tel.: +7-917-4073127

**Abstract:** The existing body of academic literature reveals that production, distribution, and consumption might be both consistently connected and geographically scattered. This requires assessing the spatial order of production–distribution–consumption cycle, within which exploring of spatial relationship would be based on mutual dependence on each other's of production, distribution and consumption. Hierarchical and spatial nesting of production, distribution, and consumption data allows us to apply hierarchical spatial autoregressive models (HSAR). The study was conducted on data from 2132 municipalities within 84 regions of the Russian Federation in 2018. The created models enabled distinguishing intraregional and interregional effects and highlighted the positive effect of spatial interactions in production volume. The calculations showed that population income, which determine the demand for goods are positively associated with production volume while relationship between manufacturing and wholesale is negative, resulting in revision of relations between wholesale and manufacturing enterprises and boosting ways of improvement the competitiveness of manufactured goods. The results allow us not only to enhance understanding of the spatial pattern of production–distribution–consumption cycle, but also to reveal new opportunities in the development of supply chain location policy.

**Keywords:** production–distribution–consumption cycle; spatial order; hierarchical linear model; hierarchical spatial autoregressive models

**JEL Classification:** R12; R15; C21

## 1. Introduction

One of the fundamental economic cycles, which ensure territorial development is the production–distribution–consumption cycle, which is understood as the simplest form of the value chain (VC) and describes the full range of activities, which are required to bring a product or service from conception, through the different phases of production, delivery to final consumers (Kaplinsky 2013). Each element of the cycle is complexly organized. The development of production, distribution, and consumption depends on many factors (Borkowski et al. 2008) that also reshape the spatial order of the whole production–distribution–consumption cycle. On the one hand, goods are produced to be consumed and it is better if production is located near the place of consumption. Demand is the main factor for the production and sale of goods (Lavoie and Stockhammer 2013) and domestic demand-led growth plays a decisive role in developing countries (Palley 2002). However, on the other side, production cannot always be located near the consumer. Special conditions are necessary for production, or the properties of certain good types, often require the commence proceedings in

areas that are distant from the places of people's residence. In turn, people often choose a place of residence based on their subjective understanding about how it should be, without associating it with the proximity to the places of goods production that causes a territorial discontinuity. Often there is a gap not only in space but also in time. This is especially true for agriculture, where the harvesting takes place during certain periods of the year (Ge et al. 2019). Distributors and wholesalers help bridge these gaps. They ensure the movement of goods produced in some territories to other ones, therefore, organizing interregional and intraregional flows (Tian and Zhang 2019). As a result, both producers and buyers gain positive effects (Zokaei and Hines 2007; Cook et al. 2011; Sadigh et al. 2013; Doan 2020). Each enterprise (not only manufacturers but the wholesalers too) taking into account general costs and given distance to consumers determines where the best place to be located. Thus, the location policy is vital for each enterprise of the production–distribution–consumption cycle (Bogataj and Bogataj 2001; Cook et al. 2011; Ge et al. 2019; Fan and Liu 2020). Such studies are of interest not only to enterprises but also to the state because they are providing better organization of the economic space, taking into account the needs of the population and enterprises and strengthening the competitive territorial advantages.

The matters of spatial organization of production–distribution–consumption cycles are closely connected with the targets of territorial administration. Given the inputs–outputs–outcomes model (Huggins et al. 2013) underlying regional competitiveness the final phase results expressed in goods range and volume could be considered as one of regional competitiveness measures embodying the population' standard of living. The broader consideration of the issue of regional disparities in the production–distribution–consumption cycle is overlapping matters attributed to growing cohesion and integration at all levels, which are sparking the interest of scientists across the world (Pietrzykowski 2019). The problem of increasing differences between the most developed and a lagged regions is often arisen across regional studies (Smętkowski 2018; Psycharis et al. 2019) including the spatial dependence (Vida and Dudás 2017). Many countries feature the concentration of highly skilled individuals and the creation technologically advanced environment within individual territories that makes them the most productivity-enhancing regions (Le Gallo and Kamarianakis 2010; Laskowska and Dańska-Borsiak 2016; Dai et al. 2017; Psycharis et al. 2019). Thus, the persistence of productivity disparities between regions points out that regional policy is still needed as an instrument of territorial cohesion (Maza et al. 2009) and balanced development (Psycharis et al. 2019).

However, existing studies are focused on examining the territorial location of production, distribution, and consumption separately. The study of the spatial organization of production, distribution, and consumption processes does not concern relationship within the cycle itself. While in many countries there is a specific cycle structure, for example, in Russia, flows of goods have historically been centralized.

This study aims to assess the spatial order of the production–distribution–consumption cycle. Our study contributes to the growing bulk of literature in two ways:

1. From the descriptive point in terms of the spatial organization of cycles and inherent gaps in the phases of production in time and space, we consider the peculiarities of cycles' centralization through the Herfindahl and Moran indices.
2. Keeping in mind the nested data, we apply hierarchical analysis methods to study the influence of demand-driving population income, wholesale trade providing distribution in regions on production.

The central point is that the distributive trade (wholesale trade) links production and consumption localized in different places, greatly identifying the production–distribution–consumption cycle. At the same time, the activities of distribution centers, wholesalers are not attributed to the local territorial scale, but as a rule, encompass a group of territories thus it requires the application of multilevel analysis methods. At the same time, the production–distribution–consumption cycle is not spatially closed; it depends on how this cycle proceeds in neighboring territories. It is possible to bear in mind both the cyclical sequence of the production–distribution–consumption processes and the spatial factor

in the framework of the hierarchical spatial autoregressive models (HSAR). This approach would allow for a better exploration of the spatial order of the production–distribution–consumption cycle.

The rest of the paper is organized as follows. The literature review embraces the approaches on spatial order of production, distribution, and consumption studying, the opportunities of examining it as linked by flows process are revealed.

Then the methods of assessing spatial autocorrelation and modeling of their relationship are presented. For better understanding, the spatial patterns of the production–distribution–consumption cycle within territory under consideration the analysis of spatial concentration, dispersion, and spatial autocorrelation of activities in Russian Federation is conducted. Based on spatial dependencies, the hierarchical model was created that both allowing taking into consideration spatial heterogeneity and spatial dependencies in the production–distribution–consumption cycle. The findings and proposals are encapsulated at the end of the article.

## 2. Literature Review

The relationship between production and consumption has been studied for many years while the approaches within which scientists conducted their study are different. Supply chain researches examine the relationship between production and consumption, but it primarily aims at finding the best ways for the movement of goods from producer to consumer. They provide an analysis of the activities related to the entire network of manufacturers, suppliers, distribution centers, wholesalers, retailers, transportation, information, and other logistics management service providers and customers. Research may involve the supply chains of individual enterprises or global supply chains whereas their goal is to find drivers on competitive advantage and ways of reducing costs and improving the efficiency of manufacturing companies (Sadigh et al. 2013; Doan 2020).

However, the movement of goods from producers to consumers is considered not only in supply chain research. The spatial aspect of the movement of goods is explored within the theoretical framework of the traditional urban theory/theory of central places. From the basic work of Lösch, the scientists repeatedly raised issues of spatial order of production, consumption, and distribution with highlighting areas of economic activities (King 2020). Its adherents consider trade center hierarchy (Borchert and Adams 1963), the role of central places in the context of flows (Philbrick 2005; Hesse 2010). Philbrick (2005) distinguished several areal units of the organization, including consumer (establishment, a group of parcels), retail (focal place, a cluster of establishments), wholesale (cluster of focal places). Borchert and Adams (1963) delineated six levels of centers: minimum convenience centers, full convenience centers, partial shopping centers, complete shopping, secondary and primary wholesale–retail. Hesse (2010) considered the system of urban places in the system of chains and highlighted the tributary areas. The rationale for their study relies on the fact that wholesale and retail distribution are closely connected with urban places where goods transshipment is treated as the classical function of the city and central places, and as gateways for providing goods and services to more distant hinterlands. In turn, the interregional movement of goods determines territorial ties. The differences in production capabilities are manifested in both positive and negative interrelationship of territories (Capello 2009). These dependencies are under active consideration within methods of spatial statistics, which make it possible to evaluate the spatial autocorrelation of indicators characterizing production (Dai et al. 2017), distribution (Hylton and Ross 2017; Tian and Zhang 2019), and consumption (Khushi et al. 2020).

The examining of spatial relationships raises the importance of modifiable areal unit problems (MAUP) due to the necessity of choosing the initial territorial units for analysis and hierarchical nature of economic space. Related studies being conducted at the macro level (Ayouba and Gallo 2019; Laskowska and Dańska-Borsiak 2016), at the micro level (Khushi et al. 2020) and at both ones (Mendes Resende et al. 2018; He et al. 2017). The scientists point out different magnitudes for coefficients at the regional иmunicipal levels that according their opinion represents the evidence of

some form of spatial interaction within the regions. Being aggregated at regional level data many interactions lose strength and are no longer statistically significant (Mendes Resende et al. 2018).

The production–distribution–consumption cycle is actively studied within the framework of the multiplier product matrix concept. In these works, the flows between economic sectors and regions are analyzed using input–output tables (Stevens and Trainer 1980; Hewings and Sonis 2016). In general, these studies are aimed at a structural analysis of the economy, where manufacturing, wholesale, and retail trade are distinguished among the sectors. However, in some cases, the concept helps to study supply chain shifting, as a result of the macro production relocation between regions (Fan and Liu 2020).

The above-mentioned works revealed two aspects that are important for subsequent analysis. First, scientists point out that data is hierarchical, but in each case, different aspects of multi-layering may be highlighted. In supply chain studies, scientists assume that coordinating all parts of the supply chain is a multilevel process (Sadigh et al. 2013) and the hierarchy of the production network emerged from the self-organization of the supply chain, which allows us to highlight the hierarchical supply chain (Kichikawa et al. 2019). Hesse (2010) argues that gateways may play a distinct yet no longer dominant role in the hierarchy of centers within the framework of the theory of central places. Building the multiplier product matrix, Hewings and Sonis (2016) identified the idea rank-size hierarchy of spatial production cycles (hierarchical feedback loop analysis).

Given the analysis of data on the volume of production, distribution, and consumption is carried out in all case studies this data could be considered as hierarchically structured. The emphasis on territorial or process multi-level ties can be made depending on the target of the study. Hierarchical analysis methods might be used for hierarchically structured data.

The second point evolved by scientists is important for subsequent analysis; it is associated with the territorial movement of goods. Across supply chain researches, product transportation distance is pivotal as it determines costs businesses (Bravo and Vidal 2013). However, another aspect is studied not so much to the distance but the nature of the relationship between manufacturers, distributors, suppliers, customers, which reflects their ability to interact to jointly address issues (Lawson et al. 2019) and form clusters (Hylton and Ross 2017). It is often vital to understanding whether distributors, suppliers, wholesalers positively or negatively affect the ability of manufacturers to sell manufactured goods and how well do they perform their functions (Tian and Zhang 2019).

Existing studies showed that the distributive trade (wholesale and retail trade) had a strong positive impact on the regional productivity growth (Di Berardino et al. 2017), while the idea that distributive trade could promote provincial convergence is viewed not considering the location of these territories relative to each other that might be done by applying contiguity matrix.

The distance is also analyzed across the developments of the central place theory, but better focused research, addresses the distribution of functions, including those related to the promotion of goods from central places to more distant hinterlands (Hesse 2010). Thus, the analysis of the central place targets assessing its ability to perform the function of promoting goods to foreign markets, and fulfilling the local market with goods produced in external ones, including neighboring territories. The multiplier product matrix makes us able to analyze the volume of goods moved from one regional system to another while identifying the dependence of one on another (Stevens and Trainer 1980; Hewings and Sonis 2016). This analysis better enables examining economic ties while excepting the mutual location of the territories. It makes possible to distinguish regions with intensive goods streams, characterizing territorial economic relationship. These territories might not be neighboring; there are other reasons could take toll such as historical ties, the need for resources etc. In its turn, spatial econometrics focuses on the mutual location of the territorial units, their geographic proximity, and distance between them. The spatial autocorrelation of production (Dai et al. 2017; Pietrzykowski 2019), distribution (Hylton and Ross 2017; Tian and Zhang 2019), and consumption (Khushi et al. 2020) are basic points of treatment. In particular, these studies show that indicators characterizing production in one territory depend on the development of production in neighboring territories (Villaverde and Maza 2008;

Fabregat and Badia-Miró 2014; Aguilar-Retureta 2016; Díez-Minguela et al. 2018; Lolayekar and Mukhopadhyay 2019; Gunawan et al. 2019). This feature manifests itself in both trade processes and consumption.

Thus, the issue of spatial relationships has arisen in various areas of research on the production–distribution–consumption cycle. This issue is essential, especially in open economies, because, it is necessary to take into account the influence of the production–distribution–consumption cycle of neighboring territories.

The interest in these explorations is caused by the target of regional disparities equalizing largely generated by the different competitiveness of regions (Vida and Dudás 2017) and regional productivity disparities (Le Gallo and Kamarianakis 2010; Smętkowski 2018). The differences in the levels of economic development between the most advanced regions and the less well-off ones still require careful consideration for the formulation and implementation of appropriate policies for a territorially balanced development (Psycharis et al. 2019).

In this study, the author examines the relationships between production and consumption in the context of territories, and with distributors and wholesalers operating at the level of a group of territories. Additionally, the impact of neighboring territories on these relationships is taken into consideration. Thus, the following hypotheses are tested by involving methods of hierarchical and spatial analysis:

**Hypotheses 1 (H1).** *The next spatial dependencies can be distinguished in each phase of the production–distribution–consumption cycle.*

Within this hypothese we assess the spatial autocorrelation of the production (i.e., how the volume of shipped goods in a certain municipality depends on the volume of shipped goods in neighboring municipalities), in what way the distribution (i.e. the volume of wholesale trade) in one region is associated with the volume of wholesale trade in neighboring regions and how population income (i.e., the population income in the certain municipality) is spatially connected with the population income in neighboring municipalities.

**Hypotheses 2 (H2).** *Production in municipalities is not determined only by the capabilities of the municipalities themselves, but also by the characteristics of the production–distribution–consumption cycle of the regions they are included in.*

**Hypotheses 3 (H3).** *Population income determines the demand for goods and have a positive effect on the volume of goods produced in the municipality.*

**Hypotheses 4 (H4).** *The activity of distribution centers and wholesalers at the level of several municipalities located in the region has a positive effect on the volume of goods produced in municipalities.*

**Hypotheses 5 (H5).** *Neighboring regions have a positive impact on the production of municipalities.*

## 3. Data of Research

Analysis of the data is being carried out on data across 2132 municipalities attributed to 84 regions of the Russian Federation in 2018. Production data are not presented for all municipalities. There is no data on closed cities, as well as on individual small areas to ensure compliance of keeping the confidentiality of primary statistical data received from organizations under the provisions of Federal State Statistics Service.

The characteristics of the indicators are presented in Table 1, Figures 1 and 2. To focus on the relationship between production, distribution, and consumption of goods all indicators had been converted per capita for not taking into account the population of the areas under consideration. The data is presented in rubles, the official currency of Russia. The ruble/US dollar exchange rate in 2018 was within a fairly wide range from 55.7 ruble/US dollar to 69.99 ruble/dollar.

**Table 1.** Description of variables.

| Indicator | Characteristic | Number of Observations |
|---|---|---|
| Municipalities (Level 1, lower) | | |
| Production ($P_{ik}$) | Production is characterized by the volume of goods produced and shipped, works and services of the next industries: mining, manufacturing, provision of electricity, gas, and steam, water (not including small businesses, whose share in the industrial production of the Russian Federation is less than 5%) | 2132 |
| Population income ($I_{ik}$) | Population income reflects the volume of social transfers and the population's taxable income | 2319 |
| Regions (Level 2, upper) | | |
| Distribution ($D_k$) | Distribution characterizes activities of distribution centers, wholesalers, suppliers, and other participants distributing goods from manufacturers to retailers and customers, it is reflected in wholesale turnover. According to the methodology of the federal state statistics service, wholesale turnover include the volume of activity that is not observed by direct statistical methods while taking into account organizations whose wholesale trade is the main type of economic activity | 84 |
| GRP per capita (gross regional product) | GRP as generalized indicator of the region economic activity, it represents the gross value added created by residents of the region | 84 |

Source: Federal state statistics service (https://www.gks.ru/dbscripts/munst/), the Unified Interdepartmental Statistical Information System (https://www.fedstat.ru/).

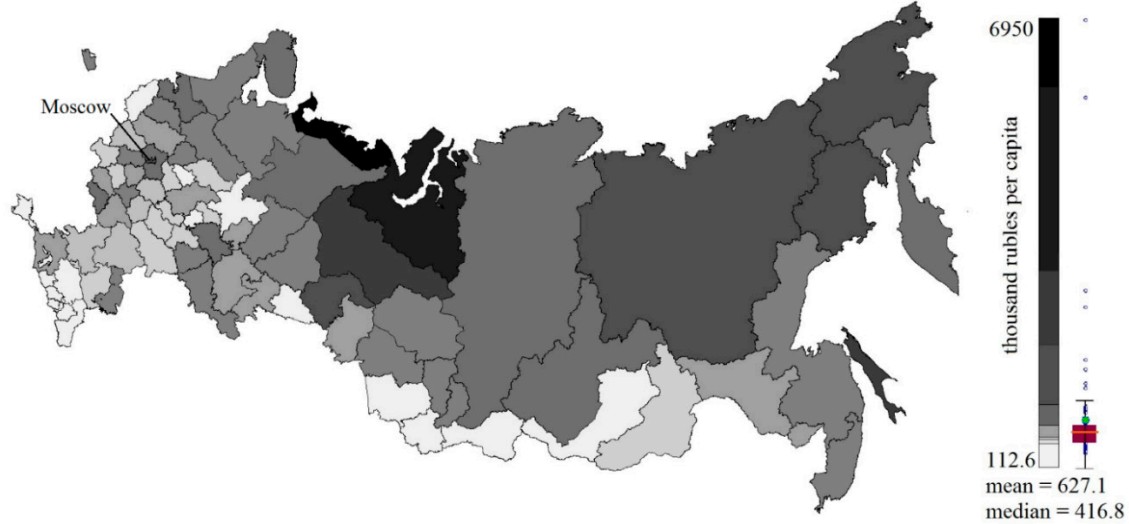

**Figure 1.** Gross regional product (GRP) per capita in the Russian Federation in 2018. Source: elaborated on the basis of the Unified Interdepartmental Statistical Information System (https://www.fedstat.ru/).

This study seeks for the assessment of spatial effects at the regional level, where an adjacency matrix (W) was used, which takes into consideration the neighborhood of first-order territories. The following assumptions were made while the adjacency matrix is being created: the Sakhalin region is considered as adjacent to the Primorsky, Khabarovsk, and Kamchatka territories, despite the water barrier; the Kaliningrad region is considered as adjacent to the city of St. Petersburg and the Smolensk region. The latter assumption is controversial, but it allows us to capture the Kaliningrad region in the study and consider the entire territory of the country as a whole, taking into account the fact of sea traffic and rail link between these territories.

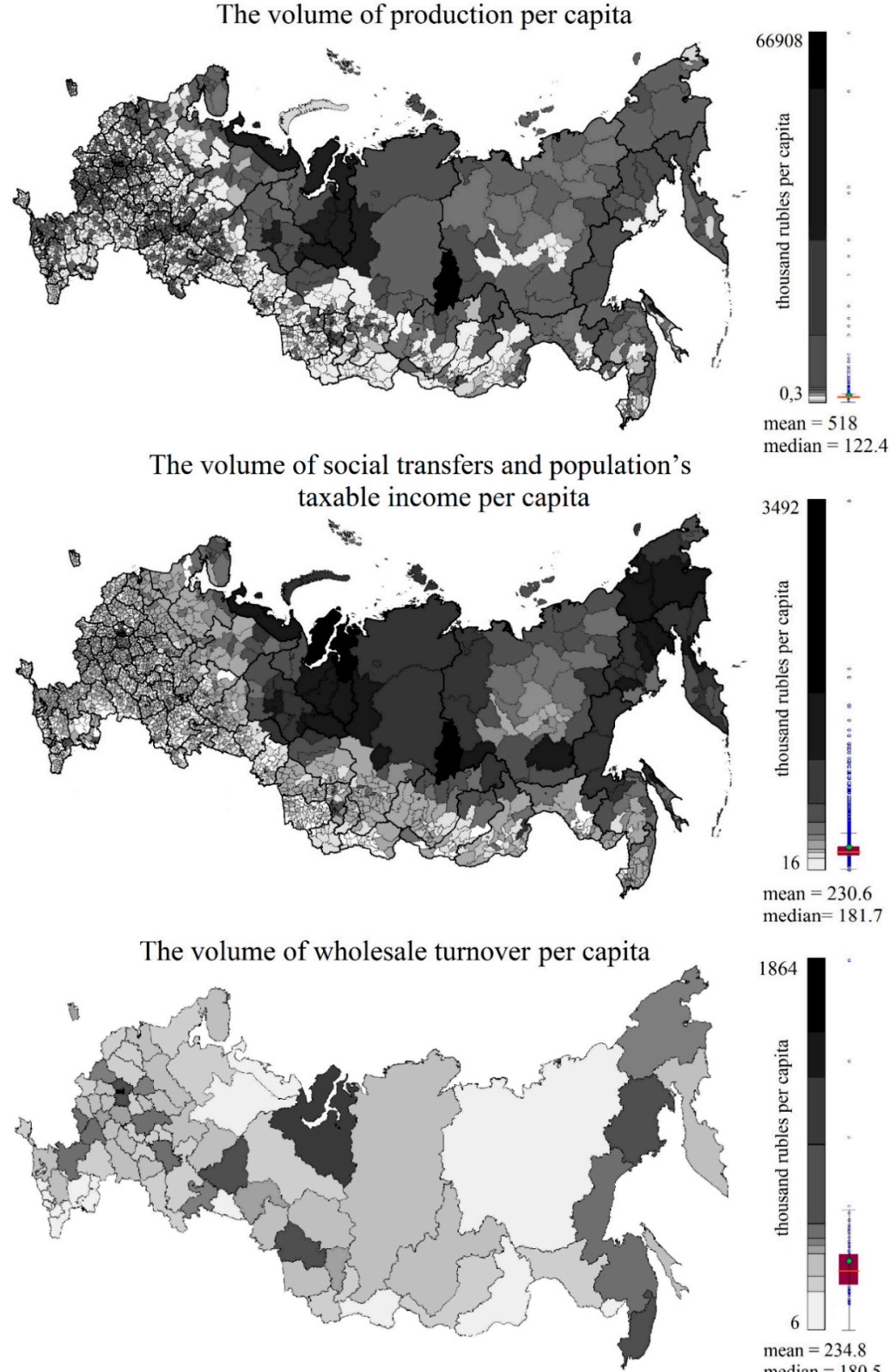

**Figure 2.** Spatial order of production, population income, and distribution of Russia in 2018. Source: elaborated on the basis of the Federal state statistics service, the Unified Interdepartmental Statistical Information System.

## 4. Methodology

The data describing the production–distribution–consumption cycle is hierarchically structured, i.e., nested. Manufacturers produce goods, the volume of shipped goods by manufacturers in total accounts for the volume of goods shipped in the municipality. In turn, the sum of goods produced in municipalities is the volume of goods shipped in the region. Thanks to the supply chain network, this volume of shipped goods in the region is sold to consumers in the municipalities in which they are produced or to the outside market. In supply chain distribution centers and wholesalers combine the flows of goods produced by manufacturers and not consumed in municipal entities, thus they organize intra-regional (inter-municipal) and inter-regional flow of goods. In this case, we observe one of the functional roles of the center, stated within the framework of the central places theory (Figure 3).

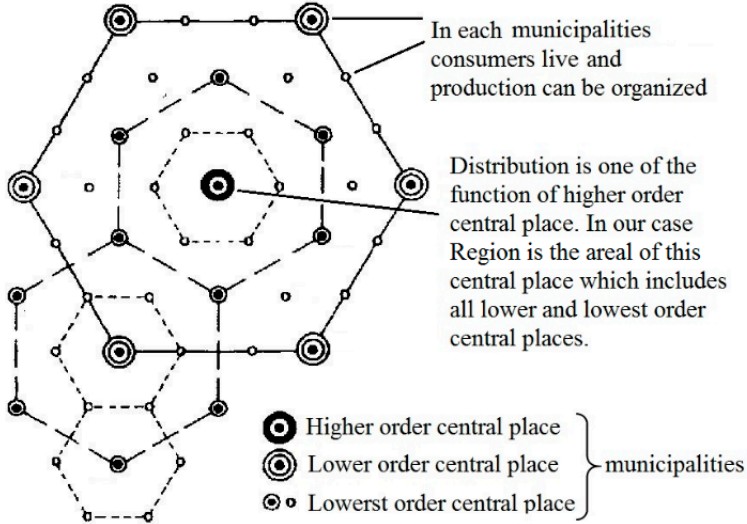

**Figure 3.** Hierarchy of central place. Source: adapted from Christaller (1933), and King (2020).

The research methodology consists of several steps. Firstly, concentration, dispersion, and spatial autocorrelation of production, distribution, and the population income, which determine the volume of consumption, are estimated.

The interaction of neighboring territories while the mutual influence of territories on each other studied using the methods of spatial statistics. The estimated spatial correlation is an assessment of the dependence of the indicator values attributed to certain territory on the values of the neighbors. This approach had been applied for the first time in the middle of the XX century in the works of Moran (1948) and Geary (1954), still it is currently used to study dependences in production (Dai et al. 2017; Pietrzykowski 2019), distribution (Hylton and Ross 2017; Tian and Zhang 2019), and consumption (Khushi et al. 2020). Global Moran's index gives us a summative evaluation of territorial connectivity, Equation (1):

$$Im_{kh} = \frac{N}{\sum_i \sum_j w_j} \frac{\sum_i \sum_j w_{ij}(x_i - \bar{x})(x_j - \bar{x})}{\sum_i (x_i - \bar{x})^2}, \tag{1}$$

where $i, j$—indexes used to label territories ($i = 1 \ldots N, j = 1 \ldots N$);

$N$—the number of examined territories, units;

$\bar{x}$—the average value of indicator;

$w_{ij}$—the contiguity matrix.

Moran's index evaluates spatial dependence (autocorrelation) of spatial data; it shows the degree of the linear relationship between the vector of indicator value the territory under consideration and the vector of spatially weighted values of the same indicator in neighboring territorial units. The Moran's

index value is compared with the expected value E (I) = −1/(n − 1). If the value of the index is higher than expected there is a positive spatial autocorrelation, the values of observations in neighboring territories are similar. Otherwise, negative autocorrelation is observed; the values of observations in neighboring territories differ. In the case when the value of the Moran's index is equal to the expected one, it is considered that the values of observations in neighboring territories are randomly distributed.

The existing researches demonstrate that spatially organized data could be hierarchically structured simultaneously (Car and Frank 1994; Timpf and Frank 1997). Thus, the second step of our study is assessing the hierarchical link of the production–distribution–consumption cycle via hierarchical linear modeling (HLM). This method was originally introduced to study group and intergroup differences in human behavior (Goldstein 2010; Garson 2013; Raudenbush et al. 2011). Over time, however, the ability to identifying group effects has been used to analyze production (Lee 2018; Yusupov et al. 2018), consumption (Chen 2012), and firm reaction speed (Lawson et al. 2019) within the supply chain.

Within the framework of this study, the HLM method enables to separate production, distribution, and consumption, taking into account their territorial coverage and spatial ties. Therefore, the wholesale trade turnover of the region depends on how much goods are produced and purchased in the municipalities included in it. In addition, the HLM method allows assessing how effects in municipalities (lower level) depend on the region (upper level) in which they are located. To that, the intraclass correlation coefficient (*ICC*) is calculated, Equation (2):

$$ICC = \frac{\tau^2}{\tau^2 + \sigma^2} \tag{2}$$

where $\sigma$, $\tau$—intra-group variance of municipalities (the within-group variance) and regions (the between-group variance).

This equation represents the ratio of between region (upper level) variance to the total variance. The coefficient value varies in the range from +1, where the variance is determined directly by the difference between groups (regions) in the absence of variance within the groups, to 1/(N − 1), where the variance is predominantly intra-group (where N—number of territories from lower level). The coefficient value near zero shows that the upper level (for example, regions) does not make effects on production volume in lower level territories (municipalities).

On the third step, we combine spatial and hierarchical analysis to study production–distribution–consumption cycle. A hierarchical analysis is based on the alternative way of capturing spatial effects, permitting to focus on the multi-level aspects of causal relationships (Corrado and Fingleton 2011) thus it becomes possible a modified contiguity matrix to be included in hierarchical models (Cellmer et al. 2019). Applying hierarchical spatial autoregressive models enables us to take into consideration both heterogeneity and hierarchical structure of data simultaneously with the identification of group and spatial effects (Cellmer et al. 2019).

Several models were consistently built in the work. At the first step, a zero hierarchical linear model (HLM) was constructed—Equations (3) and (4):

Level 1 (lower, municipalities):

$$P_{ik} = \beta_{0k} + r_{ik} \tag{3}$$

Level 2 (upper, regions):

$$\beta_{0k} = \gamma_{00} + u_{0k} \tag{4}$$

where $P_{ik}$—the volume of production per capita in $i$-th municipality attributed to $k$-th region, thousand rubles/per capita;

$\beta_{0k}$—a function of a general intercept ($\gamma_{00}$) for all municipalities, and error of interregional variance ($u_{0k}$) that characterizes variations across regions due to values of indices belonging to their municipalities;

$r_{ik}$—random error associated with $i$-municipality in region $k$.

$k$—index for affiliation of a municipality to region ($k = 1, 2, \ldots, 84$);

$i$—index for affiliation to a particular municipality ($i = 1, 2, \ldots, 2132$)

In addition to the group effect, the estimation of spatial effect might be included, thus, a zero HSAR model could be obtained, Equations (5)–(7):

Level 1 (lower, municipalities):

$$P_{ik} = \beta_{0k} + r_{ik} \tag{5}$$

Level 2 (upper, regions):

$$\beta_{0k} = \gamma_{00} + b_0 \tag{6}$$

Spatial dependence:

$$b_0 = \lambda W b_0 + u_0 \tag{7}$$

where $\lambda$—parameters of spatial interactions;

$W$—spatial weight matrix at the upper level (1 if neighbors are with common boundary and 0 if another);

$b_0$—vector of random effects for the absolute term;

$u_0$—vector of random effects.

The consistent inclusion of the factors at the lower and upper levels allows us to analyze the contribution of each of them, Equations (8)–(10).

Level 1 (lower, municipalities):

$$P_{ik} = \beta_{0k} + \beta_{1k} I_{ik} + r_{ik} \tag{8}$$

Level 2 (upper, regions):

$$\beta_{0k} = \gamma_{00} + \gamma_{01} D_k + b_0 \tag{9}$$

Spatial Dependence:

$$b_0 = \lambda W b_0 + u_0 \tag{10}$$

where $\beta_{1k}$—regression coefficient associated with I;

$\gamma_{01}$—regression coefficient associated with D relative to Level-1 intercept;

$I_{ik}$—population income in $i$-th municipality related to $k$-th region, thousand rubles/per capita;

$D_k$—indicator, characterizing of distribution in $k$-th region.

If $\lambda = 0$ then it will be full HLM model.

The proposed HSAR model assesses the municipality's production (P) dependence from population income (I), the activity of distribution centers and wholesalers in the region of municipality location (D), and the development of neighboring regions.

The variance ($r^2$) may be used to calculate a measure of effect size:

(a) explained by the Level-1 predictor variable in the outcome variable, Equation (11):

$$r_\sigma^2 = \frac{\sigma^2_{null} - \sigma^2_{random}}{\sigma^2_{null}} \tag{11}$$

(b) explained variance in the outcome variable, by the Level-2 predictor variable, Equation (12):

$$r_\tau^2 = \frac{\tau^2_{null} - \tau^2_{random}}{\sigma^2_{null}} \tag{12}$$

As an overall test of whether the regression model with predictors is a significantly better fit than the intercept-only (null) model without predictors can be used the likelihood ratio test. To assess improvement in model fit we examine the residual variance component, reliability estimate was calculated. HSAR models had been estimated with special comparison test.

## 5. Results

*5.1. Spatial Concentration of Production in the Russian Federation*

While analyzing spatial patterns of the production–distribution–consumption cycle it is necessary to understand and take into account the specificities of the Russian economy. It is worth noting that there is a fairly high concentration of activities in the Russian Federation. The most developed industries are the extraction of minerals, the provision of electricity and gas, which are being carried out mainly in the northern regions of the country, where the main natural resources are located. Here, given the difficult north working conditions and high returns of resources, the population receives a higher wage. Thus, in Russia regional inequality in aggregate productivity is firstly related to variability in the industry mix across regions.

By all means, across the world, we witness quite different types of the economy, but without addressing the activities we could distinguish the weak productivity industry in each country. In Russia, the high productivity industry is mining in the north. The situation is different in China, where manufacturing industries are more productive in the east of the country (Dai et al. 2017).

European countries have also regional inequality in aggregate productivity as a result of industry mix across (Le Gallo and Kamarianakis 2010), economic specialization, urban hierarchy and spatial scattering of population (Psycharis et al. 2019). Moreover, some regions being more productive than others, due to some aggregate factors. For instance, agglomeration processes and concentration of human capital are leading to an increase in productivity (Le Gallo and Kamarianakis 2010; Laskowska and Dańska-Borsiak 2016). Territories with the concentration of highly skilled individuals and the technologically advanced environment are becoming more competitive, reinforcing the regional inequalities (Psycharis et al. 2019) Researches reveal that tendency of growing differences is more pronounced when comparing the development of on core (metropolitan) and non-core (non-metropolitan) regions (Smętkowski 2018; Psycharis et al. 2019).

In Russia, these processes are taking place as in many European countries. The bulk of the people live in the south-west of the country, which is characterized by a rather strong differentiation of economic development between the administrative centers of the regions and the rest of the municipalities included in them. Large cities have a higher concentration of human capital, which contributes to increased innovation and benefit from economies of scale.

Gross regional product (GRP per capita) is the basic indicator characterizing productivity. Figure 1 shows that only some regions have high values while calculation of concentration and dispersion for analyzing regional disparities demonstrates spatial dependence of concentration indicators from each other. To capture this Moran's index is applied.

Calculation of the Moran's index showed that spatial autocorrelation is observed for all four examined indicators. For international comparisons, it is best to consider GRP per capita, since the calculation of this indicator is universal for all. Research shows that Moran's index for GRP is varying over a wide range (Appendix A). Both, the applied weighting matrix and the level of data aggregation (municipality, region) are crucial for calculation (He et al. 2017). The modifiable areal unit problem is arisen by many scholars (Nelson and Brewer 2015; Mendes Resende et al. 2018). They note that in analyzing of aggregate data, some interactions loses strength (Mendes Resende et al. 2018), resulting in the gained estimates of spatial dependence might be distinct (He et al. 2017).

The ranges of Moran's indexes for GDP are not wide in the Russian Federation. In the period 1996–2006, the values were from 0.37 to 0.39, in a further Moran's index dropped to 0.28 in 2011, which is largely due to the crisis 2008. Then since Moran's indexes for GDP per capita had increased to 0.36 in 2018 (Figure 4). Statistical data on of gross domestic product are not acquired across municipalities in Russia. To examine spatial relationships in details we calculated the indicator of the volume of goods, works, and services produced and shipped («Production» further in the text) which covers the results of production activities in municipalities. Data presented in Table 2 and Figure 2 demonstrates a high concentration of production. When comparing data on GDP per capita at the regional level

(Figure 1) and data on production per capita at the level of municipalities (Figure 2) we can conclude that, in general, the concentration points coincide.

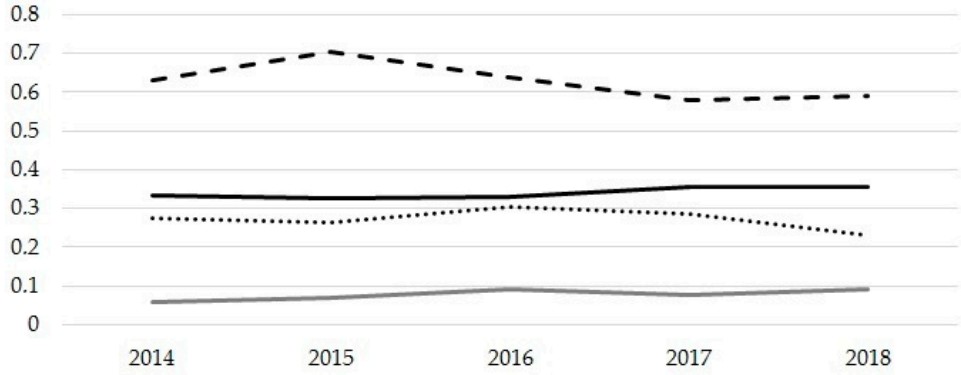

**Figure 4.** Spatial order of production, population income, and distribution of the Russian Federation in 2018. Source: own calculations using data from Federal state statistics service, the Unified Interdepartmental Statistical Information System.

**Table 2.** Indicators of concentration and dispersion of activities in the Russian Federation in 2018.

| Indicators | Regions Level (Upper) | | Municipalities Level (Lower) | |
| --- | --- | --- | --- | --- |
| | GRP | Distribution | Production | Population Income |
| CV (coefficient of variation) | 2.06 | 3.61 | 16.7 | 10.6 |
| Gini | 0.65 | 0.77 | 0.91 | 0.8 |
| HHI (Herfindahl–Hirschman index) | 0.06 | 0.16 | 0.13 | 0.05 |
| Atkinson index | 0.31 | 0.52 | 0.77 | 0.57 |
| Rosenbluth index | 0.03 | 0.05 | 0.005 | 0.002 |

For the assessment where used the aggregate values of the regions which weighted by population. Source: own calculations using data from Federal state statistics service, the Unified Interdepartmental Statistical Information System.

At the same time at the municipality level, we observe that high Moran's indexes in regions are explained by the influence of certain municipalities. High indicators of concentration and dispersion on «Production» refer to the high variation of the values within regions. If in the northern regions the differences between municipalities are associated with disparities in the extraction of natural resources, then in the southwestern part of the country these differences are determined by the concentration of activity in large cities, with inadequate development of production in the peripheral municipalities of the regions. Bearing that the production Moran's index at municipal level is below in comparison with the Moran's index related to GDP per capita we could assume that observed concentration and dispersion could negatively affect the spatial dependence within regions. Thus, the data at the municipal level provide more detailed information about the spatial organization of economic activity. This connected with the modifiable areal unit problem, under which dependencies that were revealed at micro level could be not detected on aggregated data (Mendes Resende et al. 2018).

Moran's index positive in terms of production (0.23–0.31), but it is not so high as in terms of population income, where Moran's index during the period 2014–2018 ranged from 0.58 to 0.7. Such autocorrelation is visually manifested in a smooth transition of color from darker to light in Figure 3. The positive Moran's index shows that a high population income in some municipalities is observed against the background of a relatively high population income in neighboring municipalities. Conversely, low-income municipalities tend to be adjacent to areas within which the population

has similarly low income. The lowest value of the Moran's index is attributed to wholesale trade, which characterizes distribution. In 2014–2018, it varied in the range of 0.06–0.09.

Table 2 shows that distribution is more concentrated, thus the functions performed by distribution towards production seem to be interesting. Hypothesis H1 is confirmed, and spatial dependencies could be distinguished in the production–distribution–consumption cycle. At the same time, estimates of concentration and dispersion of activities require an in-depth analysis on the relationship between production, distribution and consumption.

### 5.2. Results of Modeling

The constructed zero HLM model (Model 1) allows us to test the hypothesis H2 that production in municipalities being determined not only by the capabilities of the ones but also by the characteristics of the production–distribution–consumption cycle of the regions involved. The ICC shows that 15.8% of the variance in production (P) is at the group level and 84.2% of it is at the individual municipal level (Table 3).

**Table 3.** Estimated results of the models.

| Variables | HLM | | | HSAR | |
|---|---|---|---|---|---|
| | Model 1 | Model 2 | Model 3 | Model 4 | Model 5 |
| Intercept, $\gamma_{00}$ | 643.7 ** (138.9) | −2049.3 ** (516.87) | −1611.9 ** (357.2) | 707.8 * (289.9) | −1630.3 ** (294.3) |
| Population income, $\beta_1$ | | 10.6 ** (2.2) | 10.8 ** (2.3) | | 10.9 ** (0.3) |
| Distribution, $\gamma_{01}$ | | | −2.2 * (1.0) | | −2.1 ** (0.5) |
| Spatial interactions, $\lambda$ | | | | 0.61 | 0.72 |
| ICC | 0.158 | 0.157 | 0.151 | 0.12 | 0.09 |
| Reliability estimate | 0.784 | 0.782 | 0.775 | 0.781 | 0.766 |
| The value of the log-likelihood function | −19,797.2 | −19,196.6 | −19,189.1 | −19,798.4 | −19,186.4 |
| Deviance | 39,594.4 | 38,393.2 | 38,378.2 | 39,596.7 | 38,372.98 |
| Regular HLM vs. HLM with spatial dependence model comparison test | | | | $\chi^2$ statistic = 11.2 ** | $\chi^2$ statistic = 18.67 ** |

** *p*-value < 0.001 * *p*-value < 0.05, the values of standard error are within brackets. Source: own calculations using data from Federal state statistics service, the Unified Interdepartmental Statistical Information System.

Further, Model 2 was built including the independent variable at Level 1.

The final estimation of fixed effects was evaluated with robust standard errors, which showed that population income (I) has a positive effect on the change in production (P). In this case, the population income determines its demand for goods, thus, the higher the demand for goods, the greater the volume of products shipped in the municipality. However, when interpreting the results, another relationship should be noted: high volumes of production per capita allow paying higher wages, which in turn determines a higher level of consumption. At the same time, $r^2$ calculated after the construction of Model 2 indicates that the variable of population income explains 36.2% of the variance in production at Level 1, thus a decrease in ICC to 15.7%, deviance reducing from 39,594.4 to 38,393.2 indicates a decline in the unexplained part of the dependent variable's variance. Thus, hypothesis H3 is confirmed.

Model 3 includes a predictor at Level 2. Estimates indicate that distribution (D) has a negative impact on production (P) that contradicts the proposed hypothesis, H4. This hypothesis was in line with the existing assumption about distribution centers and wholesalers within the supply chain should promote goods produced in one territory both to domestic and foreign markets, thereby enhancing the effects of manufacturers. At the same time, aside from a strong focus on efficiency improvements, consumer focus is also present in the context of the supply chain. Scholars point out that in the competitive markets, the supply chain works as an interrelating network of suppliers, manufacturers,

distributors, and customers, to satisfy customer demands. With this in mind, taking into account the interests of consumers, distributors and wholesalers can purchase goods on foreign markets. With domestic producers unable to compete with ones from other regions, these supplies negatively impact their own sales opportunities. This happens in Russia while the country has highly extractive industries with low competitiveness in the production of goods for final consumption, which are mainly supplied from abroad in processed forms. Thus, hypothesis H4 is rejected, which is due to the peculiarities of the domestic market of the Russian Federation. At the same time, after the construction of Model 3 calculated $r^2$ indicates that distribution explains 6% of the variance in production at Level 1. Both decrease in ICC to 15.1% and the value of deviance to 38,378.2 indicates a small unexplained part reduction of the dependent variable variance.

Models 4 and 5 confirm the H5 hypothesis that neighboring regions have a positive impact on the municipalities' volume of production. As a result of adding the spatial interactions into the model, ICC decreased to 9% while deviance rose to 38,372.98. This indicates that HSAR models better describe the production–distribution–consumption cycle in the region.

Model 4 has no predictors and could be compared with Model 1. The result of the comparison test provides evidence that the HSAR provides a better fit, as indicated by the $\chi^2$ statistic of 11.2, df = 1, $p = 0.001$. A comparison of the standard errors for $\gamma_{00}$ the regular HLM and HSAR (138.9 vs. 289.9) suggests that, that there is an underestimation of the standard errors when spatial dependence is ignored. Model 5 shows the population's income has a positive impact on production while the impact of distribution is negative. The development of the production–distribution–consumption cycle in neighboring territories has a positive effect on production volumes in municipalities.

## 6. Conclusions

The study was aimed at assessing the spatial order of the production–distribution–consumption cycle. It deals with the several aspects at once, including the study of supply chain, which focuses on finding the ways for the movement of goods from producer to consumer and theory of central places that allows to join the system of urban places with the system of chains and to highlight the tributary areas. Our findings allowed the justifying the presence of hierarchical and spatial relationships in the production–distribution–consumption cycle the investigation of which covers estimation of concentration, dispersion, spatial autocorrelation and the hierarchical dependence of activities. The need of simultaneously analyzing these relationships, on the one hand, is determined by the complexity of the production–distribution–consumption cycle, and on the other hand, by its importance in the economy of regions and the country in general. The broader consideration of production–distribution–consumption cycle revealed its connectivity with regional competiteness and spatial disparities.

Taking into account the modifiable areal unit problem our analysis was based on two levels of aggregated data, which are a region and a municipality. Our findings illustrate high spatial concentration and dispersion of activities in the Russian Federation. Despite this, there is a spatial autocorrelation of production, estimated using the Moran's index per capita. The range of Moran's index change is not so great in comparison with other countries, indicating that there are no notable changes both towards integration and gap in spatial relationships across the Russian Federation. At the same time, there are differences in estimates carried out at the regional and municipal levels. Regionally aggregated data do not manifest all interactions. Thus, the lower spatial autocorrelation arisen from the calculation of the Moran's index for the production indicator at the municipal level indicates weak intraregional relationships.

The calculations also showed that spatial interactions are sufficiently strong for the population income and weak for the indicator characterizing the distribution phase in the production–distribution–consumption cycle.

The nested, hierarchically and spatially structured data made it possible to apply hierarchical spatial autoregressive models. The advantage of such models is their ability to separate intraregional and interregional effects in the production–distribution–consumption cycle, taking into account the

existing spatial ties. The calculations showed that the 15.8% of variance in production in Russia attributed to region within which a municipality is located. The constructed models pointed out that the population income have a positive effect on production whereas the distribution has a negative effect on it. Greatly, the negative influence of distribution is associated with the peculiarities of the Russian economy; it shows that the activities of distribution centers and wholesalers make a small contribution to the promotion of the produced goods on the markets; these activities are aimed at obtaining effects by consumers.

This situation took place due to the lengthy period of a command-type administration of the economy in which one ministry determined what would be produced in the country and where the products would be supplied. Recent three decades witnessed the absence of sustainable patterns of production–distribution–consumption cycles in majority Russian regions while wholesale companies located in the capital of the country account for almost 60% of the total distribution. That situation cannot but influence the regional competiveness that is explained by bottom-up approach (Camagni 2002) in which enterprises placed on initial hierarchical level utilizing existing resources and a capacity of the environment are launching the production and able to generate the knowledge and innovations (Huggins et al. 2013). By all means, the distribution channels weakness of manufactured products is not a key problem, however, studies show that the weak development of the distribution system (for example, in terms of exports (Camagni 2002)) negatively affects the overall competitiveness of the territory. Thus, in combination with poor industrial development and very modest results of the ongoing reforms aimed at diminishing of interregional asymmetry, the existing imbalances in economic development persist and negatively affect the competitiveness of regions and the country as a whole.

The obtained estimates of the negative relationship between production and distribution require deeper study. It is advisable to separate data on the movement of materials and consumer goods, the information on the mining and manufacturing industries; in the context of consumer groups goods (clothing, food products, electrical equipment, etc.), their specificities can also be highlighted. To deepen the understanding of problem there the share of highly value added industries should be distinguished in total production structure that could highlight additional growth points of territorial competiveness. A similar separation can be made for distributors and wholesalers. Another limitation of the study is that the territorial neighborhood was taken into account only at the regional level. At the same time, the conducted analysis showed that the spatial autocorrelation of production is more pronounced at the level of municipalities. Accordingly, future research may be aimed at finding methods to take into account spatial dependencies at both levels of the administrative-territorial division.

At the same time, the constructed models made it possible to confirm the hypothesis of positive spatial interactions within the production–distribution–consumption cycle, that is to say the growth of the production and the consumption, and the progress of wholesale in regions are positively spatially associated with neighboring territories manifesting the cohesion in the country. Thus, while defining the location policy of manufacturers, distributors and, wholesalers in the region it is necessary to take into account not only the needs of the region but also sales opportunities, development of production and distribution in neighboring territories.

**Author Contributions:** Conceptualization, V.T.; methodology, V.T.; software, V.T.; validation, V.T., K.G. and D.K.; formal analysis, K.G.; investigation, V.T. and D.K.; resources, V.T. and K.G.; data curation, V.T.; writing—original draft preparation, V.T.; writing—review and editing, V.T., K.G. and D.K.; visualization, V.T.; supervision, V.T. and K.G.; project administration, D.K.; funding acquisition, V.T. All authors have read and agreed to the published version of the manuscript.

**Funding:** This research was funded by the Ministry of Science and Higher Education of the Russian Federation (scientific code FZWU-2020-0027).

**Conflicts of Interest:** The authors declare no conflict of interest. The funders had no role in the design of the study; in the collection, analyses, or interpretation of data; in the writing of the manuscript, or in the decision to publish the results.

## Appendix A

**Table A1.** Moran's indices in countries of the world.

| Variable | Country | Matrix | Moran's Index | Year | Source |
|---|---|---|---|---|---|
| GDP per capita | EU | 10-nearest neighbors the queen contiguity | 0.27–0.44 | 2000–2015 | Ayouba and Gallo (2019) |
| GDP per capita | EU | weight matrix of 2nd order | 0.496 | 2014 | Laskowska and Dańska-Borsiak (2016) |
| Regional GDP per capita | Mexico | weighted by contiguity | 0.1–0.4 | 1895–2010 | Aguilar-Retureta (2016) |
| Regional (NUTS3) per-capita GDP | Spain | weighted by contiguity | 0.12–0.64 | 1860–2010 | Díez-Minguela et al. (2018) |
| GDP per capita | Indonesia | weighted by contiguity | 0.1–0.25 | 2000–2017 | Gunawan et al. (2019) |
| Per Capita Net State Domestic Product | India | weighted by contiguity | 0.15 0.22 | 1981 2010 | Lolayekar and Mukhopadhyay (2019) |
| Relative GDP per capita | China | no data | 0.01–0.22 | 1989–2012 | Dai et al. (2017) |

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
