# Peer review of "Spatial Patterns of Production-Distribution-Consumption Cycle: The Specifics of Developing Russia"

_economies, doi:10.3390/economies8040087_

Round 1
Reviewer 1 Report
The reviewed article deals with ith an interesting problem i.e. analysis of spatial order of production, distribution and consumption cycle based on the Russian Federation case. I have no major criticism of this manuscript. Both the hypotheses, the methodology used and the description of the results do not raise any doubts. The research procedure has been precisely described and the results properly interpreted. In my opinion, the biggest weakness of the subject undertaken in the article is the structure of the economy of the Russian Federation. Taking into account that the highest productivity is characterized by the extraction of raw materials, the obtained results are burdened with the key role of the mining industry, blurring the importance of consumer goods production. It should be acknowledged, however, that the author does not mask these limitations. He openly mentions them both in the description of the results and the conclusions, properly indicating the directions of further research. Taking this into account, I consider the reviewed article as a valuable contribution and recommend it for publication.
Author Response
Dear reviewer!
First of all we would like to thank you for your review report and our article positive estimation.
Point 1: In my opinion, the biggest weakness of the subject undertaken in the article is the structure of the economy of the Russian Federation. Taking into account that the highest productivity is characterized by the extraction of raw materials, the obtained results are burdened with the key role of the mining industry, blurring the importance of consumer goods production. It should be acknowledged, however, that the author does not mask these limitations. He openly mentions them both in the description of the results and the conclusions, properly indicating the directions of further research.
Response 1: We agree that the biggest weakness of the subject undertaken in the article is the structure of the economy of the Russian Federation. We understand this shortcoming but we are restricted in available data. At the same time we expanded the discussion of the structural problems of the economy of the Russian Federation in Conclusion (line 539-553)
Besides, the additions were made in article taking into account the comments of the second reviewer. In the Introduction after the general information (importance of production-distribution-consumption cycle in economic processes ect.), after having added the information that discloses why this study is of increasing importance we showed how it is connected with both the regional cohesion and the regional development. (line 52-57, 60-65, 199-203). Also we showed how our study connected with the modifiable areal unit problem (MAUP) (line 134-141) and detailed this expression of the problem attributed to our research (line 403-406, 423-429). Walter Christaller citation was added under the Figure 3. (line 269). Inside the results part of the draft manuscript we explained assumption that regions have economic performance differences across countries (199-203, 385-389). We mention exchange rate ruble to dollar in 2018. (line 234-235). We change the 3. Methodology and 4. Data of research part of the article. A final subtotal paragraph was removed before the main summary (line 492-494). We improved the general conclusion (line 503-524, 538-552, 557-559, 566-568).
Warm regards, authors
Reviewer 2 Report
Dear Authors!
First of all I would like to thank this informative article which is dealing with "Spatial patterns of production-distribution-consumption cycle: the specifics of developing Russia"
I would like to give some recommendations to improve this manuscript.
Theoretical background and first part of the results:
The literature review part of the paper is well written, but i would like to recommend some improvement.
After the general information (importance of production-distribution-consupcion cycle in economic processes ect.), it would be necessary to explain how it relates to regional competitiveness (some theory and examples).
Another theory would be necessary to mention the modifiable areal unit problem (MAUP), because your paper partly dealing whit this.
As for me Figure 1. based on Walter Christaller has modell, it would be worth mentioning him.
Inside the results part of the draft paper (line 353 and 357) it would be necessary to explain some more informations to engage this paper in international processes. One way, regions have economic performance differences across countries (urban/rural divide, east/west divide, declining industrial areas/developing industrial areas).
So I would like to recommend some articles dealing with territorial difference, polarisation and regional competitiveness accross different countries:
Huggins, R., Izushi, H., Thompson, P. 2013. Regional Competitiveness: Theories and Methodologies for Empirical Analysis.
Camagni, R. 2002. On the Concept of Territorial Competitiveness: Sound or Misleading? Urban Studies
Psychairs, Y. & Kallorias, D. & Pantazis, P. (2019): Regional Inequalities in Central and Eastern European Countries: The Role of Capital Regions and Metropolitan Areas. In. Śliwiński, A. Polychronidou, P. Karasavvoglou, A. (Eds.), Economic Development and Financial Markets: Latest Research and Policy Insights from Central and Southeastern Europe. (pp. 3-20). Springer Nature Switzerland AG, Cham.
Smetkowski, M. (2018): The role of exogenous and endogenous factors in the growth of regions in Central and Eastern Europe: the metropolitan/non-metropolitan divide in the pre- and post-crisis era, European Planning Studies, 26(2), pp. 256-278.
Vida, G. & Dudás, G. (2017): Geographical context of the revealed competitiveness of urbanised areas in Hungary excluding the Budapest agglomeration, Geographica Pannonica, 21(3), pp. 179-190.
These studies maybe help to integrate your case study in a wider theoretical background.
Please mention that the rubel approximately how much dollar or euro, based on 2018 exchange rate. This help to the international readers, accross Europe and USA.
The methodological part of the paper is well written, but i would like to offer that change the 3. Methodology and 4. Data of research part of the article.
Data and research and after the methodology, i think this is more logical.
The results part and modells are well written, a final subtotal paragraph is required before the main summary.
The general conclusion must be improved a little bit, based on theory extensions. At the moment, discussion is too descriptive. It would be necessary to refer back to the theory better, why it is important to analyse a complexity of production-distribution-consupcion cycle and how related with competitiveness, MAUP etc.
Finally this is a good paper, and if the little suggestions and offered articles will be integrated in this article, I will recommend this manuscript to publish in this journal.
The reviewer
Author Response
Dear reviewer!
First of all, we would like to thank you for your review report which covers the interesting set of literature in our research field. To improve our manuscript we tried to meet your all recommendations.
Point 1: Theoretical background and first part of the results: The literature review part of the paper is well written, but i would like to recommend some improvement. After the general information (importance of production-distribution-consumption cycle in economic processes ect.), it would be necessary to explain how it relates to regional competitiveness (some theory and examples).
Response 1: In the Introduction after the general information (importance of production-distribution-consumption cycle in economic processes ect.), after having added the information that discloses why this study is of increasing importance we showed how it is connected with both regional cohesion and regional development. (line 52-57, 60-65, 199-203)
Point 2: Another theory would be necessary to mention the modifiable areal unit problem (MAUP), because your paper partly dealing whit this.
Response 2: We showed how our study connected with the modifiable areal unit problem (MAUP) (line 134-141) and detailed this expression of the problem attributed to our research (line 403-406, 423-429)
Point 3: As for me Figure 1. based on Walter Christaller has model, it would be worth mentioning him.
Response 3: We agree with Walter Christaller citation therefore this citation was added under Figure 1. (line 269)
Point 4: Inside the results part of the draft paper (line 353 and 357) it would be necessary to explain some more informations to engage this paper in international processes. One way, regions have economic performance differences across countries (urban/rural divide, east/west divide, declining industrial areas/developing industrial areas). So I would like to recommend some articles dealing with territorial difference, polarisation and regional competitiveness accross different countries: Huggins, R., Izushi, H., Thompson, P. 2013. Regional Competitiveness: Theories and Methodologies for Empirical Analysis.Camagni, R. 2002. On the Concept of Territorial Competitiveness: Sound or Misleading? Urban Studies Psychairs, Y. & Kallorias, D. & Pantazis, P. (2019): Regional Inequalities in Central and Eastern European Countries: The Role of Capital Regions and Metropolitan Areas. In. ÅšliwiÅ„ski, A. Polychronidou, P. Karasavvoglou, A. (Eds.), Economic Development and Financial Markets: Latest Research and Policy Insights from Central and Southeastern Europe. (pp. 3-20). Springer Nature Switzerland AG, Cham.Smetkowski, M. (2018): The role of exogenous and endogenous factors in the growth of regions in Central and Eastern Europe: the metropolitan/non-metropolitan divide in the pre- and post-crisis era, European Planning Studies, 26(2), pp. 256-278.Vida, G. & Dudás, G. (2017): Geographical context of the revealed competitiveness of urbanised areas in Hungary excluding the Budapest agglomeration, Geographica Pannonica, 21(3), pp. 179-190.These studies maybe help to integrate your case study in a wider theoretical background.
Response 4: Inside the results part of the draft manuscript, we explained the assumption that regions have economic performance differences across countries by using kindly proposed articles (199-203, 385-389)
Point 5: Please mention that the rubel approximately how much dollar or euro, based on 2018 exchange rate. This help to the international readers, accross Europe and USA.
Response 5: We mentioned the exchange rate ruble to the dollar in 2018. (line 234-235)
Point 6: The methodological part of the paper is well written, but i would like to offer that change the 3. Methodology and 4. Data of research part of the article. Data and research and after the methodology, i think this is more logical.
Response 6: We changed the 3. Methodology and 4. Data on the research part of the article.
Point 7: The results part and modells are well written, a final subtotal paragraph is required before the main summary. (line 492-494)
Response 7: A final subtotal paragraph was removed before the main summary.
Point 8: The general conclusion must be improved a little bit, based on theory extensions. At the moment, discussion is too descriptive. It would be necessary to refer back to the theory better, why it is important to analyze a complexity of production-distribution-consumption cycle and how related with competitiveness, MAUP etc.
Response 8: We improved the general conclusion: the theoretical review was generalized to demonstrate its influence on methodology; we expanded it and showed how gained results are consistent with the other studies (line 503-524, 538-552, 557-559, 566-568).
We hope that we understood all your comments in the right way, we appreciate your insightful comments and now our article looks better.
Warm regards, authors